# The COVID-19 Health Crisis and Its Impact on China's International Relations

**Jean-Pierre Cabestan**

Department of Government and International Studies, Hong Kong Baptist University, Kowloon, Hong Kong, China; cabestan@hkbu.edu.hk

**Abstract:** Using qualitative methods, this article focuses on the relationship between the COVID-19 health crisis and China's foreign policy and foreign relations. My main argument is that since its outbreak in late 2019, the COVID-19 health crisis has deepened the tensions already existing between China and the United States, as well as China and the West in general. Other factors that appeared before the pandemic have also contributed to intensifying the Sino-US rivalry as well as Sino-European frictions. Nonetheless, Beijing's proactive mask and vaccine diplomacy, its strict lockdown policy as well as its more aggressive nationalist and anti-western narrative have fed rather than alleviated these tensions. While China's image in the Global South has remained largely positive, in the Global North, it has rapidly deteriorated. All in all, this paper demonstrates that the pandemic has been an aggravating factor contributing to the downward spiral of China's relations with the outside world as well as its own isolation.

**Keywords:** COVID-19; China; United States; European Union; Xi Jinping; Donald Trump; Joe Biden

## 1. Introduction

To what extent has the COVID-19 health crisis influenced and changed China's foreign policy and international role? To what extent has it affected China's relations with the United States and the European Union (EU)? How has it influenced China's relations with the Global South? These are the questions that this article ambitions to briefly address. These questions relate both to China's international policies and actions as well as outside perceptions of and reactions to these policies and actions.

To be sure, many of China's policies and actions (e.g., its assertiveness and competition with the US (and the West)) predate the pandemic. Nonetheless, my hypothesis is that the pandemic has highlighted and enhanced China's assertiveness as well as intensified this competition. Instead of fostering cooperation as some, including the Chinese government, had hoped (*Xinhua* 2020b), COVID-19 has fed frictions and tensions between China and developed democracies. At the same time, COVID-19 has contributed to increasing China's influence in the Global South in spite of several cases of pushback which the West has managed to capitalize upon.

Not many articles have been published on this issue. I contributed a study in the early phase of the crisis (Cabestan 2020a). Among the other analyses, some have questioned or minimized the impact of the pandemic on Chinese foreign policy (Raman and Murkherjee 2021) and more broadly the world order (Nye 2020). As we will see, I do not completely disagree with this view, although I do think that COVID-19 aggravated the situation. Aside from that, works that speculated that China would use the pandemic to assert its ambition to change international norms have made the right prediction (Haenle 2020).

This article adopts a neo-realist approach of international relations inspired by Randall Schweller (2006). While it recognizes the structural nature of the Sino-US geo-strategic rivalry, it also takes into account the influence of domestic factors, particularly of the political and ideological differences between both powers' polities. Yet, it also factors in

perceptions and misperceptions, to use a concept promoted by Robert Jervis (1976), and how nationalism has strengthened their impact on policies.

Using qualitative methods, this article attempts to isolate the pandemic factor and assess its weight and impact on China's foreign policy and foreign relations. While it covers a short period of time, which started in late 2019 after the new coronavirus was discovered in Wuhan, it also puts back the emergence of the pandemic in the broader context of growing competition between China and the United States and China and the West in general. This article is divided into four sections: the first one briefly presents China's growing international ambitions prior to the outbreak of the pandemic; the second one analyzes China's COVID-19 diplomacy and its "overkill" dimension; the third one evaluates the impact of COVID-19 Chinese activism on its relations with the north, especially the United States and the European Union; and the final section assesses the impact of China's COVID-19 diplomacy on its relations with the Global South.

## 2. China's International Role before COVID-19

To better isolate COVID-19 and assess its impact, it is necessary to briefly revisit and present China's growing international role before the pandemic erupted in late 2019 and became a pandemic in the following spring. Since Xi Jinping became General Secretary of the Chinese Communist Party (CCP) in late 2012 and before 2020, China had already more clearly asserted its foreign policy posture. As a result, its relationship with the United States became more contentious, and with the European Union, it became more complicated.

### 2.1. A More Assertive Foreign and Security Policy

It can be argued that this assertiveness goes back to 2008, the Beijing Olympics and the international financial crisis. Then, the Chinese government started to feel that the West, and particularly the United States, were in decline, opening wider China's window of strategic opportunity. Yet, it was Xi, not his predecessor Hu Jintao, who abandoned for good Deng Xiaoping's low-profile diplomacy (韜光養晦 *daoguang yanghui*) and replaced it with a much more proactive and assertive foreign policy, encapsulated in the slogan "strives for achievements" (奮發有為 *fenfa youwei*) (Doshi 2021a).

There was no coincidence that it was around the same time in the autumn of 2013 that Xi announced the launching of the new land (one belt) and maritime (one road) silk roads, or "one belt, one road" (OBOR, 一帶一路 *yidai yilu*), a policy that was 2 years later renamed in English the "Belt and Road Initiative" (BRI). Aimed at intensifying China's economic connectivity with and as a result their footprint in the world, the BRI mainly enhanced its loans and investments in infrastructure projects in the Global South (Zhao 2020).

Since 2012 as well, Beijing has started to become more assertive in the maritime domain that it claims and within what is called the "first island chain", a string of islands which goes all the way from Okinawa to Borneo via Taiwan and the Philippines. In the South China Sea, after having annexed Scarborough Shoal in April 2012, a land feature long occupied by Manila and located 200 km west of Luzon, the Chinese government started to build artificial islands on the rocks that it had controlled since the late 1980s in the Spratlys and later militarized them (Hayton 2014; Shambaugh 2020). In the East China Sea, Chinese coast guard ships increased their intrusions in the contiguous and territorial waters around the Senkaku (or Diaoyu), a group of small islands administered by Japan since 1895 and claimed by China and Taiwan since the early 1970s (O'Hanlon 2019).

That aside, after Ms. Tsai Ing-wen's election as president of Taiwan and the return of the independence-leaning Democratic Progressive Party (DPP) to power, Xi has adopted a much more aggressive posture toward the island state, indicating a clear intention to not only refuse any contact with the new Taiwanese authorities as long as they do not endorse the "one China principle" but also to speed up unification with the mainland by any means, including the use of force and the implementation of the "one country, two systems" formula in the island (Cole 2020).

This more ambitious foreign and security policy was confirmed by Xi in his report to the 19th CCP Congress in 2017. Now, China is "moving towards the world's central stage" and hopes to become a full-fledged military great power by 2049, the occasion of the 100th anniversary of the People's Republic of China, capable of supplanting the United States, including militarily, and as a result replacing it as the world's top superpower (Xi 2017).

In order to achieve these new objectives, the Chinese government has introduced a more proactive but also aggressive foreign policy posture, often referred to as "Wolf Warrior diplomacy" (戰狼外交 *zhanlang waijiao*). This expression was inspired by the successful 2017 Chinese movie titled *Wolf Warrior 2*, which describes how China is now able to protect its interests and nationals overseas, especially in civil war-torn African countries (Martin 2021).

### 2.2. A Growing Rivalry with the US

China's new objectives and ambitions could only trigger a more intense rivalry with the United States. While president Obama had already decided to "pivot" to Asia as early as 2011, the US rebalancing became more decisive under his successor Donald Trump. In the meantime, introducing in 2015 its "Made in China 2025" strategy, Beijing had shown a clear intention to catch up with America on the technological front. In the same period, spurred by the BRI, US-China competition had intensified in the Global South, in Southeast Asia, in Africa, in the Middle East, in Latin America and even in the South Pacific (Breslin 2021). In the same period of time, the People's Liberation Army (PLA) sped up its modernization, particularly the expansion of its navy. Additionally, in 2017, it opened its first naval base overseas in Djibouti, a strategic chokepoint situated in the Horn of Africa and next to the Bab-el-Mandeb and the Gulf of Aden (Cabestan 2020b).

The launching of the US-China trade war in the spring of 2018 has been the most visible and publicized feature of this growing rivalry. However, far from being only commercial, this competition has also rapidly appeared to be both geostrategic and ideological. In 2019, the Trump administration adopted a new Indo-Pacific strategy openly aimed at better balancing, or some would say "containing", China's growing diplomatic and military ambitions in the region. At the same time, the Quad, the quadrilateral security dialogue among the US, Japan, India and Australia, was revived and strengthened. Although the Trump Administration did not initially give much attention to human rights issues, the deterioration of the situation in Hong Kong after the failure of the 2019 protest movement and in Xinjiang, where more than 1 million Uighurs had been detained without trial, forced it to react, and this was before the COVID pandemic broke out (Pillsbury 2020).

### 2.3. A More Complicated Relationship with the European Union

Prior to COVID, EU-China relations had already become more difficult. In March 2019, the EU Commission had stated that "China is, simultaneously, in different policy areas, a cooperation partner with whom the EU has closely aligned objectives, a negotiating partner with whom the EU needs to find a balance of interests, an economic competitor in the pursuit of technological leadership, and a systemic rival promoting alternative models of governance" (European Commission 2019). The number of irritants had obviously increased, including the lack of market access of EU companies in China, the forced technology transfers imposed on them when they invest in China, the subsidies provided by the Chinese government to its companies, especially its state-owned enterprises (SOEs), in spite of its commitments to the WTO rules, Beijing's promotion of 16 + 1 (17 after Greece joined in 2019 and 16 again after Lithuania left in 2021), a forum bringing Central and Eastern countries together every year, including 12 (now 11) EU member-states, and China and promoting business and investment relations with the help of the BRI, as well as human rights issues, a bone of contention which had become more serious as the Xi government intensified its crackdowns of dissidents, activists and autonomy defenders in Tibet, Xinjiang and later in Hong Kong (Godement 2020).

### 3. China's COVID-19 Diplomacy: Overkill

When the health crisis turned into a pandemic, China became hyperactive on the international stage, strongly defending its own behavior and strategy (*Xinhua* 2020b). However, it ambitioned to achieve too many objectives at the same time, both trying to appear as a generous country and manifesting an offensive nationalism. As a result, Beijing's "overkill" boomeranged, feeding the existing tensions and contributing to isolating China (Cabestan 2020a).

#### 3.1. China's Defensive and Offensive Strategies

On the one hand, the Chinese authorities pursued defensive objectives. The coronavirus originated in Wuhan in December 2019, but until 23 January 2020, when Xi Jinping decided to execute a national lockdown, they tried to hide the existence of a pandemic and let its citizens travel abroad, facilitating the dissemination of the virus. In contrast, after 23 January, China's behavior changed 180 degrees; it closed the country from the outside and imposed strict quarantine measures on all travelers.

At the same time, Beijing embarked on proactive COVID diplomacy, providing large quantities of masks and PPE (personnel protective equipment) to a large number of countries both in the Global North and the Global South. However, that was not all. To better defend its own policy, the Chinese government minimized the assistance that it first received from other countries (e.g., the EU) and launched a disinformation campaign about the origin of the virus, propagating the idea that some American army athletes had transmitted it to Chinese counterparts in the international military games organized in Wuhan in October 2019 (Cabestan 2020a).

On the other hand, the Chinese government adopted an offensive strategy, trying hard to appear as a responsible great power and a model member of the international community, in particular to the World Health Organization (WHO) (*Xinhua* 2020a). It promoted the idea that China was a "model" in terms of health crisis management, capitalizing on the difficulties encountered by other countries, particularly democracies, and boasting about the efficiency of its CCP-led system of governance. More generally, the Chinese propaganda apparatus took advantage of the COVID crisis to intensify its nationalist discourse, with the intention to both glue the society around the government and sell China's success story around the world (Zhao 2022). In the WHO as well, it worked hard to be perceived as a model participant in the global fight against the pandemic, and this was despite an attempt to manipulate the organization leadership as well as a reluctance to provide timely information to the WHO and let WHO experts investigate in China, especially in Wuhan. For example, Beijing put pressure on WHO Director General Tedros Adhanom Ghebreyesus to postpone until 30 January its decision to declare COVID-19 a public health emergency of international concern (PHEIC) and until 11 March to elevate it to the status of pandemic, slowing down any global or country-based measures to stop it.

In other words, spurred by a growing nationalism at home that they stimulated on purpose, the Chinese authorities have attempted to reach too many objectives at the same time, compromising the efficiency of both their message and their action.

#### 3.2. The Impact of China's COVID Strategy on Its Relations with the World

Rapidly, criticism of China's behavior vis-à-vis the pandemic emerged in the world, first in the developed countries and also in the Global South.

Beijing's criticism of the Trump administration's early decision, made in late January 2020, to stop air traffic with China did not go over well in the United States, all the more because, not long after, the Chinese authorities did the same and in a more forceful manner. However, it was the latter's attempt to control the WHO's COVID discourse and strategy that triggered the US' fiercest criticism. A former Ethiopian Health Minister, Tedros was himself attacked for being too close and obedient to China, which took advantage of this inclination as well as the opacity of its own health system to impose its own approach on the WHO (Feldwisch-Drentrup 2020). Since the WHO is a rather weak multilateral

organization which is traditionally highly dependent upon the cooperation of the country where the epidemy originates from, the Chinese government easily achieved its objective. For example, Beijing managed to ban worldwide any reference to the "Wuhan flu" and instead call the new pandemic COVID-19.

Moreover, China's lack of transparency and limits imposed on any WHO investigation team fed suspicions about the veracity of the data that it publicized. Officially, only around 130,000 Chinese people got COVID, and less than 5000 of them subsequently died of COVID (130,398 and 4824, respectively, by 25 February 2022) (BBC 2022). However, reports published in the US in 2020 already indicated much higher figures—probably up to 2.9 million cases and tens of thousands of deaths—even if these data show that China managed to keep the virus under control much faster than many other countries (Scissors 2020). More recent studies have made Chinese official data even less credible (Calhoun 2022a, 2022b).

Another criticism had to do with the quality of the equipment delivered by China and the fact that the Chinese government mixed up aid with sales, blurring the distinction between the two and propagating the false impression that it was more generous than any other nation (ChinaPower 2021).

More generally, China's Wolf Warrior diplomacy and anti-western discourse have led to a battle of narratives with many countries (Sun 2020). This has included the European Union and, in particular, France. There, the Chinese Ambassador Lu Shaye was summoned to the French Foreign Ministry in April 2020 for having disseminated on the Embassy's website inaccurate information about the lack of attention given to COVID patients in some French hospitals (Huang 2021).

A more unexpected development took place around the same time in Canton (Guangzhou), where cases of racial discrimination emerged; a number of Africans had been expelled from their flats, their landlord fearing that they were contaminated with COVID. This triggered strong reactions from the public opinion and even the government in several African countries, especially Nigeria, forcing the Chinese Foreign Ministry to go out of its way to mend relations with them (Ngcobo 2021).

Another issue that the pandemic has exacerbated has been developing countries' growing debt. To be sure, far from all their debt is due to China. In June 2020, Beijing endorsed the debt relief plan known as the "G20 Debt Service Suspension Initiative" (DSSI) adopted by the G20 meeting held that month. Then, it announced the suspension of debt repayment for 77 developing countries, as well as a financial help of USD 2 billion over 2 years for countries hit by the coronavirus pandemic (*Global Times* 2020). Later however, the Chinese government clearly showed a preference for rescheduling debt repayments rather than granting debt forgiveness, affecting its image in the Global South and particularly in Africa (see below).

The final question was the fierce competition for vaccines' R&D and international distribution. Instead of mitigating tensions among great powers, this competition has intensified them. China offered its own emergency vaccines to many developing countries even before they had been approved by the WHO, triggering suspicion about their safety and efficacy among the recipient countries. Later, Beijing joined the WHO's sponsored COVAX distribution scheme, competing with the US and Europe to appear as a key player in vaccine donations (see below).

All in all, China's COVID activism and nationalism have had mixed results. While its mask, PPE and vaccine diplomacy may have consolidated China's position in some developing countries, overall, it has contributed to deteriorating its image and underscoring a growing rivalry with the West and not only the United States.

## 4. China's Post-Pandemic International Activism and Changing Perception of China

China's deteriorating image in the West and part of the Global South has not stopped its ambition to play a bigger role on the international stage. On the contrary, its ability to get out of the pandemic and to resume normal economic activities as soon as in the second

half of 2020 and consequently ahead of most countries convinced the Beijing authorities that they could take advantage of this new situation to push their envelope and become even more self-confident and proactive.

As a result, they intensified their anti-Western discourse and diplomacy. While it started to be mentioned in 2017, "the East is rising the West is declining" (東升西降 *donsheng xijiang*) as a slogan became much more quoted both by the CCP propaganda and Chinese international relations experts (Doshi 2021b). This new discourse has been more successful in the Global South than in the "Global North".

*4.1. China's Post-Pandemic International Activism*

First of all, since it approved its first homemade vaccine (Sinopharm) in December 2020, China has developed proactive vaccine diplomacy, especially in poor countries. There, it is a major player today. In May 2021, Sinopharm became the first Chinese vaccine endorsed by the WHO. This was followed a month later by CoronaVac, manufactured by Sinovac. As a result, in October 2021, both Chinese vaccines counted for almost half of all the COVID-19 vaccines distributed globally, with CoronaVac being the most frequently used vaccine in the world today ahead of Pfiser-BioNTech (Mallapaty 2021). True, their efficacy may be lower (79% and 51%, respectively, against over 90% for BioNTech). Nonetheless, in early August 2021, China claimed to have already supplied many more doses (750 million against 110 million for the US) to developing countries than others (104 against 65) (Leng and Hu 2021). These recipients included a lot of BRI nations.

This distribution effort, however, has been far from free. Among the 1.3 billion doses distributed by China by early October 2021, an estimated 71.9 million doses, or 5.5%, had been donated. True, China was ahead of the US, which announced only in May 2021 the distribution of 80 million doses free of charge worldwide. However, the perception that China was selling rather than donating vaccines compelled the Chinese government to adjust its policy. In September 2021, Xi Jinping pledged that China would donate 100 million COVID-19 vaccine doses to developing countries by the end of this year (Ma 2021). In November 2021, at the 8th Forum of China Africa Cooperation (FOCAC) in Dakar, Xi promised to provide 1 billion doses of vaccines to Africa, including 600 million doses as a donation (*Xinhua* 2021a). Still, 40% of the Chinese vaccines distributed to Africa will be sold rather than donated.

The other issue for which China also adjusted (to some extent) its policy is debt relief and rescheduling. In 2020–2021, Chinese lenders provided USD 12.1 billion in global debt relief, including USD 1.3 billion under the G20 DSSI relief program. However, by July 2021, the amount of interest-free loan debts cancelled remained small—USD 114 million to 15 African countries—against repayment deferrals amounting to around USD 7 billion, mainly in favor of Angola (USD 6.2 billion) and Ecuador (USD 891 million) (China Africa Research Initiative (CARI) 2021).

Beyond COVID-19 and the debt crisis, China wants more than ever to be seen both as the leader of the South and the world's "silent majority", providing a new and better model of governance to the world. To that end, Beijing has continued to invest in the United Nations system, enhancing its influence in it and convincing several of its agencies to adopt Xi Jinping's formula about the "common future of mankind". More broadly, the Chinese Communist Party has intensified its ideological rivalry with the West, promoting its own political system as "a democracy that works" in opposition to American and more generally Western democracies which are, in its eyes, unable to deliver good and efficient governance (*Xinhua* 2021b).

*4.2. China's Contrasted Image in the World*

China's post-COVID-19 international activism has had a contrasting impact. While its image has clearly deteriorated in the North, it has overall remained strong in the Global South.

In October 2020, according to the Pew Center Survey (Silver et al. 2020), unfavorable views of China had reached "historic highs" in many countries. Across the 14 nations surveyed, a median of 61% thought that China had done a bad job dealing with the COVID-19 outbreak. This was many more than those who said the same of the way the pandemic had been handled by their own country, by the WHO or by the EU (except the US (84%)), Japan (79%), Australia (73%), France (54%) or Italy (49%).

In June 2021, while US's image had recovered (61% favorable view), China's image has remained largely negative among developed countries (27% favorable view) (Silver 2021).

The limit of these surveys is obvious: only developed countries were included.

If we look at Afrobarometer data for example, China's image may have gone down a bit compared to pre-COVID times, but favorable views about this country's influence in Africa remained largely positive (60% in 2020 against 65% in 2015). This does not mean the Africans' view of the US has deteriorated; it has remained rather strong (58%) ahead of the former colonial powers (46%) or Russia (38%). However, rather than opposing both great powers, as many developing countries' citizens, Africans that "feel positively about the influence of China are more likely to hold positive view of the US influence" (Sheehy and Asunka 2021).

Yet, in closing its borders for over 2 years, China complicated its relations with many countries, especially the Asian part of the world, which has close economic relations with it (Wang 2021; Yang 2022).

## 5. COVID-19 Pandemic Impact on China-Global North Relations

China's activism both after the outbreak of the pandemic and in the post-pandemic period has had a very negative impact on its relations with the Global North. The United States under Trump, and later under Biden, has adopted a much more confrontational strategy toward China. While the EU has been slower to adjust, its China policy has also evolved toward a more critical and robust posture.

### 5.1. Impact on China-US Relations

By and large, the COVID-19 pandemic has contributed to deepening the Sino-US rivalry. More visible at the end of the Trump presidency, this rivalry has intensified under Biden. Of course, as we will see, other factors have played a role. Among them, the repression of the 2019 protest movement and the introduction of a National Security Law in June 2020 in Hong Kong need to be mentioned. These developments convinced Secretary of State Mike Pompeo to push later that year for a regime change in China. However, China's more assertive posture has united the Americans in their intention to push back. Consequently, the Biden administration has largely carried on Trump's China policy. The trade war has gone on, even if the original intention to "decouple" with the Chinese economy has been abandoned. In September 2021, together with Australia and the United Kingdom, the US concluded a new pact, AUKUS, whose objective is also to rein in China's aggressive military plans, especially vis-à-vis Taiwan and in the South China Sea. Aside from that, on human rights, while Biden has become more subdued on the regime change issue, he has increased his government's pressure as well as the ideological battle between authoritarian China and democracies (Manning 2021).

Yet, the way the Chinese authorities have handled the health crisis has been an aggravating factor in the deterioration of China-US relations. For example, as early as January 2020, the Trump administration accused that Chinese leaders "intentionally concealed the severity" of the pandemic from the world. This was followed by a battle of narratives about the origins of the virus which added to the tensions between both countries (Council on Foreign Relations 2021).

With Biden, some hoped that Sino-American cooperation on the pandemic at least would take shape, but this has not been the case. After the Biden administration re-joined the WHO, both powers deepened their competition both in terms of vaccine distribution and debt relief. While the US has been slow to provide vaccines and assistance, it has been

catching up with the firm intention to not let China dominate the game. More generally, the pandemic has accelerated the US plan to reorganize supply chains and reduce its dependence upon China and not only as far as masks, PPEs or vaccines are concerned.

All in all, the pandemic has consolidated a US-China geo-strategic rivalry. Since this rivalry is structural, the result has not been surprising (deLisle 2021).

### 5.2. Impact on China-EU Relations

The EU and EU member states have been slower to react. The growing dissatisfaction of European public opinion with China did not prevent the EU from negotiating and approving in principle the Comprehensive Accord on Investment (CAI) with China in December 2020. Here again, other factors, especially the increasingly worrisome human rights situation in Xinjiang and Hong Kong, have played their part in the deterioration of Sino-European relations. For the first time in its history, in March 2021, the EU imposed targeted sanctions on a few Chinese officials in charge of Xinjiang, triggering Beijing's decision to launch a series of countersanctions against a few European activists, including members of the EU Parliament who had been critical of China. This latter decision has contributed to compromising any endorsement of the CAI by the EU Parliament, leaving it in limbo in the foreseeable future.

Nonetheless, China's handling of the pandemic has accelerated the deterioration of Sino-EU relations. Meanwhile, in spite of COVID and the growing Sino-US tensions, some EU countries such as Hungary or non-EU nations such as Serbia have maintained a good relationship with China, and spurred by a more critical public opinion, the EU mainstream view about China has changed (Seaman 2020). Less naiveté and more realism were already perceptible in the EU China policy in the summer of 2020 (Le Corre and Brattberg 2020). However, more recently, the EU's attitude toward China has not been more accommodating in spite of Xi Jinping' repeated effort to drive a wedge between Europeans and Americans and US clumsiness toward France in the preparation and announcement of the AUKUS pact (Ganster 2021). As with the US, the EU wants to continue its dialogue and cooperation with China on many issues, including of course climate change (Emmott 2021), but the pandemic has helped the EU realize the true ambitions of China not only in its neighborhood, especially vis-à-vis Taiwan, but also in the world. In other words, competition has clearly taken the lead over cooperation.

### 6. Conclusions

Using qualitative methods as well as surveys, this article has demonstrated that the COVID-19 health crisis has been an aggravating factor, contributing to making both China's foreign policy more assertive and its relations with the outside world, especially its major partners, more contentious. Having gotten out of the health crisis and its domestic lockdown (with a few local exceptions) faster and having fewer cases than any other large country, China has good reason to feel stronger today. As a result, it has no reasons to change course. The successful organization of the Beijing Winter Olympics in February 2022 and the long Sino-Russian joint statement issued on the day of its opening ceremony, which President Vladimir Putin attended, are perfect illustrations of China's and Xi Jinping's self-confidence and ambition to change the world order (Yu 2022). True, the pandemic has not fundamentally changed China's foreign policy and its relations with its major partners, nor it has really changed the world order. In that sense, I agree with other analyses presented above (Nye 2020; Raman and Murkherjee 2021). As this article has also shown, today's trends in Chinese foreign policy were already at play before. Nevertheless, the pandemic has intensified tensions and irritants, deteriorating China's image, especially in the North (except in Russia), and accelerating the readjustment of US's and later EU's own China policy. While geostrategic realities and differences of political values laid the groundwork for COVID-19 to become a source of contention (Schweller 2006), perceptions both in China and outside of China have played a role (Jervis 1976).

The pandemic has given an opportunity for China to reach out to more countries as well as enhance its cooperation with them within the WHO (COVAX) or bilaterally. Its image in the Global South has gone down a bit, but it has remained largely positive.

Taking advantage of the BRI, China's diplomatic activism has directly helped maintain this overall good perception. However, at the same time, the COVID-19 pandemic has slowed down China's interactions with the rest of the world, contributing to isolating the country from the international community. Xi Jinping has not left China since January 2020 and has met a handful of foreign official guests since then. Most other leaders, such as Prime Minister Li Keqiang, have not traveled abroad either, leaving this job to the two most senior officials in charge of foreign affairs, Politburo member Yang Jiechi and Foreign Minister (and State Councilor) Wang Yi. Overall, justified by the quarantine restrictions imposed on any large Chinese delegation traveling abroad, this deliberate decision to reduce to a trickle high-level contacts with the outside world has been detrimental to China, both for its image and the credibility of its foreign policy, and the organization of more frequent visioconferences with other heads of state or government could not really replace face-to-face meetings. The surge of new COVID-19 variants, such as Omicron, has been conducive to keeping many of China's doors closed, feeding distrust with the rest of the world, particularly the Global North.

More broadly, the COVID-19 health crisis has intensified China's competition with the Global North. Instead of bringing together nations to fight against a common evil, such as the battle against climate change arguably succeeded in doing in Glasgow in November 2021 (COP26), the pandemic has contributed to making competition prevail even more over cooperation in the context of a growing rivalry and perhaps a new Cold War between China and the US and, more broadly, China and the West.

**Funding:** This research was funded by the Hung Hin Shiu Charitable Foundation, by the Hong Kong Baptist University Research Committee (Project 165234) and by the Heinrich Boell Stiftung.

**Institutional Review Board Statement:** Not applicable.

**Informed Consent Statement:** Not applicable.

**Data Availability Statement:** Not applicable.

**Conflicts of Interest:** The authors declare no conflict of interest.

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
