# Peer review of "The COVID-19 Health Crisis and Its Impact on China’s International Relations"

_jrfm, doi:10.3390/jrfm15030123_

Round 1

Reviewer 1 Report

The author has presented his/her argument in a clear and organized manner and made relevant improvements to further add value to the paper.

Author Response

Thank you, jpc

Reviewer 2 Report

Dear Author

Your article is very interesting, and I am grateful for the opportunity to read it. I find the subject very interesting, and the results of your analysis give a lot of new information.

Reading the text, I found 3 elements that I think would improve your article.

First: The abstract I think it should clearly define the purpose of your analysis, the method, the subject, and the results. In my opinion yours is not as clear and exact as it should be. I think that the abstract structure is needed by the reader, but it is also important for you as authors - readers often use the abstract review method to search for content that interests them. You need to prepare more formal abstract – because it is not a story or newspapers highlights but information for the academics. If not, it may be a barrier to popularizing your article.

Second: Introduction - The introduction should provide not only a background but also an indication of a precise research/review objective, the research questions and methods used. Adding a brief indication of the logic of presenting the material (i.e. the brief description of the content of each section of the paper) in the last paragraph of the introduction is highly recommended.

In this case the analysis of the issue and the literature review are insufficient - a high level of generality does not allow for the presentation of the topic, the reference of the general issue to a specific analysis and presented example.

Additionally, the formal part of the article is missing. There is no description of the analysis’s scope, goal/goals, hypotheses or/and research questions. It is a necessary element of each article, thus creating a clear framework for the analysis. In general, the construction of the introduction should guide the reader and explain who, how, where, how and why the authors analyze this issue.

Third: there is no discussion. I think about the discussion as an attempt to confront your opinion with another and The Discussion as a chapter. So I miss both.

To increase the significance of the results, the discussion should embrace the differences and similarities among your findings and those of other scholars. The analysis of other studies and analyzes is clearly missing. Just as there is no reference to others points of view, different situation and examples. There is no application of a scientific analysis to reality and no real discussion. All this consequently reduces the article to the research part and causes the reader to judge the quality of the research but not its purpose or conclusions.

Fourth: The Conclusions chapter is the weak part of your article. In my opinion conclusions are insufficient - The conclusion section should be a brief summary of article’s aim, methods and findings. But it's not here. This chapter should be extended. For me, the summary is too limited, there is no reference to your opinions, assumptions, your research questions. At this point, you should show references to all formal aspects of your article. At the begging and at the end you should include a description of the research questions. Develop and explain goals. It is necessary to change the convention from the presentation of opinions to the presentation of results and conclusions. In general, I believe that research questions should be presented. The goals should be presented and explained. At the end, the conclusions should refer to each of the goals.

Summarizing.

I find your article very good. I really like your article and appreciate your work. It is interesting topic, and the conclusions could open the way for further analyses. You have to make some changes especially in Abstract, Introduction, Discussion and Conclusions. But my general opinion and my assessment of your research and whole article is more than positive.

Good luck!

Author Response

Dear Reviewer, 

Thank you for your comments and advise. I have completely followed them.

I have revised and strengthened the abstract. 

I have beefed up the introduction and included a literature review in it. 

I have updates some developments and added some reference. 

I have enlarged an strengthened the conclusion.

Thank you again for your advise. 

Best, jpc

Reviewer 3 Report

Journal: Journal of Risk and Financial Management (JFRM)

Article title: The Covid-19 Health Crisis and its Impact on China’s International Relations

General Comments:

This article studies the relations between China, the Global South and the Global North before and after Covid-19 Health Crisis. The author uses a qualitative analysis for this article. The author reached the conclusions that the Covid-19 health crisis has deepened the tensions already existing between China and the United States, as well as China and the West in general, Beijing’s proactive mask and vaccine diplomacy as well as its more aggressive nationalist and anti-Western narrative have fed these tensions. Overall, the pandemic has been an aggravating factor contributing to the deterioration of China’s relations with the outside world as well as its own isolation.

Overview:

The paper is somehow good written and the empirical work does not appears to be carefully and correctly done. The research question is mediocre and it does make a sufficient new contribution to the literature to be suitable for the Journal of Risk and Financial Management (JFRM) ONLY after MAJOR revisons.

In fact, the literature on China’s International Relations is quite vaste. The MINOR contribution of the paper is the analysis of Covid-19 health crisis and its impact on China’s international relations.

The paper is neutral interesting; and in my view, it needs to be MAJOR improved to reach the standard required for publication in this journal.

This article does NOT fit with scope and objective of Journal of Risk and Financial Management (JFRM).

The article could be transferred to other MDPI journals, more suitable to the analyzed subjects: World (https://www.mdpi.com/journal/world); Social Science (https://www.mdpi.com/journal/socsci) and Societies (https://www.mdpi.com/journal/societies).

Specific Comments:

  1. Introduction: theoretical approach + NOVELTY + results (better explanation);
  2. Literature review: is quite simple; increase with 3-4 paragraphs; Insert more ACADEMIC articles, and after 2015
  3. The model: there is no model
  4. INCREASE your analysis using some specific data
  5. The period for analysis is quite short: only the Covid-19 Health Crisis
  6. The results: need much improvements
  7. The article has one good idea: the relations between China, the Global South and the Global North before and after Covid-19 Health Crisis
  8. Conclusions: policy implications? Limitations?

General considerations: the idea of the article is not very new; the model and results must be substantially improved, and after MAJOR revisions (increase the number of the countries + period), it can be published in Journal of Risk and Financial Management (JFRM).

This article does NOT fit with scope and objective of Journal of Risk and Financial Management (JFRM).

The article could be transferred to other MDPI journals, more suitable to the analyzed subjects: World (https://www.mdpi.com/journal/world); Social Science (https://www.mdpi.com/journal/socsci) and Societies (https://www.mdpi.com/journal/societies).

Author Response

Dear Reviewer, 

Thank you for all your comments and advise. I have tried as much as possible to take them into account. But I cannot introduce major revisions since the paper has already been positively evaluated by two other reviewers who are basically satisfied with this second version provided some improvements are made in the abstract, the introduction, the literature review and the conclusion, which I did

Since it is a contribution to a special issue coordinated by a guest editor, i cannot envisage submitting this article to another journal.

I hope that you will understand.  

Best regards, 

JP Cabestan

Round 2

Reviewer 3 Report

Journal: Journal of Risk and Financial Management (JRFM)

Article title: The Covid-19 Health Crisis and Its Impact on China’s International Role and Relations.

Dear Author (s);

Dear Editor,

The manuscript has been revised  for  better  interpretations  according  to the suggestions of  the Reviewer(s),  by  including  the informations required. 

The auhor(s) change the interpretations, results and conclusions accordingly, and therefore, the paper is much improved now. The author(s) reduces considerabilly the article, references and diversify the articles cited.

I recommend that this article to be published in Journal of Risk and Financial Management (JRFM).

Congratulations!

This manuscript is a resubmission of an earlier submission. The following is a list of the peer review reports and author responses from that submission.

Round 1

Reviewer 1 Report

I read the paper with great interest. I agree with most of the arguments made by the author, especially on how Beijing's mask and vaccine diplomacy. and nationalistic and sometimes aggressive anti-Western societies narratives, along with its controversial ways of handling the Hong Kong problem and the like, have significant, if not negative, effects on how the international perceive China.

As the author brought up the discussion of Chinese nationalism, it might further add value to the paper if some discussions about "COVID nationalism" or/and "vaccine nationalism" in the world and in China can be added.

Reviewer 2 Report

The presented article, although it takes up an interesting and current topic, is not of a scientific nature. It is rather a journalistic essay, reflecting the views of the author, while taking into account the diverse literature. The following arguments prove the lack of scientific character, which in my opinion cannot be eliminated by improving the text:
- lack of a scientific structure - taking into account the description of the research problem, the adopted methodology, reference to the theoretical conditions,
- the article is not embedded in a broad literature review, as evidenced by the small number of sources used,
- no scientific language - directing the considerations towards loose journalism
- failure to specify the contribution of the article to the development of the scientific discipline,
- lack of self-conducted studies, going beyond the non-standard review of quite random literature, resulting in a conclusion unrelated to the considerations presented in the text of the manuscript.
In conclusion, I believe that the article is not suitable for publication and it is not possible to correct it. It is necessary to rewrite it from scratch by improving the structure, better and more precise definition of research problems and expanding the scientific apparatus, as indicated in the review.

Reviewer 3 Report

Already after reading the title, I'm not sure if the article covers the title, It is not clear how the author tests the impact of the Covid-19 Health Crisis on China’s International Role and Relations. The second major consideration is why the methodological approach is missing, when we test the impact, the methodological background should be clearly highlighted. I am afraid that the article is a "mathematical exercise" rather than serious research. I suggest validating the research background and adding value to your research, at this stage, given the status of a renowned impact journal, just papers with great value-added should be presented.